# Gender Differences in Adenine Diet-Induced Kidney Toxicity: The Impact of 17β-Estradiol on Renal Inflammation and Fibrosis

**DOI:** 10.3390/ijms26031358

**Published:** 2025-02-06

**Authors:** Sugyeong Ha, Minjung Son, Jeongwon Kim, Doyeon Kim, Mi-Jeong Kim, Jian Yoo, Byeong Moo Kim, Donghwan Kim, Hae Young Chung, Ki Wung Chung

**Affiliations:** 1Department of Pharmacy and Research Institute for Drug Development, College of Pharmacy, Pusan National University, Busan 46241, Republic of Korea; tnrn34@hanmail.net (S.H.); min30124@naver.com (M.S.); 98juon_k@naver.com (J.K.); kdy991117@naver.com (D.K.); yos3552@naver.com (M.-J.K.); choiceyja@naver.com (J.Y.); sjssjek717@naver.com (B.M.K.); hyjung@pusan.ac.kr (H.Y.C.); 2Functional Food Materials Research Group, Korea Food Research Institute, Wanju-gun 55365, Republic of Korea; kimd@kfri.re.kr

**Keywords:** chronic kidney disease, kidney fibrosis, inflammation, gender difference, 17β-estradiol

## Abstract

Chronic kidney disease (CKD) involves ongoing impairment of kidney function and structural changes. Previous studies indicated that males have a substantially higher prevalence of CKD than those observed in females. Here, we compared the gender differences in CKD development by comparing age-matched male and female mice subjected to a 0.25% adenine diet (AD) for two weeks. Male mice showed a significantly greater decrease in kidney function than female mice, as evidenced by the elevated blood urea nitrogen levels (M-AD: 160 ± 5 mg/dL, F-AD: 90 ± 4 mg/dL; *p* < 0.001). Furthermore, male mice kidneys exhibited pronounced tubule dilation and kidney damage, as detected by histological and biochemical methods. The extent of fibrosis was quantified using multiple biological methods, revealing a greater degree of fibrosis in male kidneys. We next indicated the inflammatory responses in the kidneys. Similar to the extent of fibrosis, AD-fed male mice showed significantly increased levels of pro-inflammatory markers, including cytokine expression and infiltration of immune cell, compared to female mice. Based on in vivo observations, the anti-inflammatory and anti-fibrotic effects of 17β-estradiol (E2) were further evaluated in vitro conditions. E2 pre-treatment significantly reduced lipopolysaccharide-induced inflammatory response through inhibition of the nuclear factor-kappa B (NF-κB) pathway in NRK52E renal epithelial cells. In NRK49F renal fibroblasts, E2 pre-treatment also reduced TGFβ-induced fibrotic responses. We further demonstrated that E2 markedly decreased fibrosis and inflammation in AD-fed mouse kidneys. Our observations revealed that male mice kidneys exhibited a heightened inflammatory and fibrotic response compared to female mice kidneys. Additionally, our findings suggest that the observed sex differences may be partially attributed to the potential anti-inflammatory and anti-fibrotic effects of E2.

## 1. Introduction

Chronic kidney disease (CKD) is a significant cause of mortality and morbidity, affecting over 800 million people worldwide by 2022 [1,2]. CKD is influenced by a combination of genetic and environmental factors that interact to determine an individual’s risk of developing the condition [1,3,4,5]. Notably, sex differences play a crucial role in determining susceptibility, disease progression, and outcome in CKD [5]. Studies have consistently indicated that men progress to end-stage renal disease (ESRD) more frequently and rapidly than women [4,6]. This disparity has been partly attributed to the influence of sex hormones. Despite these findings, the precise molecular and physiological mechanisms that drive these sex-based disparities remain poorly understood. Understanding these differences is crucial for developing sex-specific treatment strategies to improve outcomes in both men and women with CKD.

Kidney fibrosis is a central process to CKD progression and is characterized by the excessive deposition of extracellular matrix (ECM) proteins, including COL1A2, COL3A1, Vimentin, α-SMA, and Fibronectin, in the renal interstitium [7]. This process involves multiple cell types, such as fibroblasts, endothelial cells, epithelial cells, pericytes, and immune cells. A critical factor in fibrosis progression is the activation of fibroblasts, leading to their transition into myofibroblasts, which undergo proliferation and actively produce substantial amounts of ECM proteins into the interstitial space [8]. While myofibroblasts are primarily responsible for fibrosis, other cell types also play crucial roles. Pericytes, endothelial cells, and epithelial cells can transition to mesenchymal-like cell types, thereby directly contributing to fibrosis [8,9]. Additionally, epithelial cells secrete pro-fibrogenic and pro-inflammatory mediators such as TGFβ, CTGF, and various cytokines, further promoting fibrosis. Substantial evidence has indicated that inflammatory cells are critical for initiating and perpetuating renal fibrosis [10,11]. Chemokines released by injured tubular epithelial cells attract various inflammatory cells, including monocytes, dendritic cells, T cells, and fibrocytes [12]. The accumulation of these inflammatory cells is a hallmark of kidney fibrosis and contributes significantly to its progression [13].

Sex hormones, particularly androgens and estrogens, play crucial roles beyond their reproductive functions. They significantly contribute to organ development and the prevention of various disorders, including osteoporosis, colon cancer, and lung disease. Recent studies have highlighted a strong correlation between sex hormones, particularly estrogen, and the incidence of acute kidney injury [14,15,16]. Estrogen exerts its effects through multiple pathophysiological pathways linked to hormone-dependent conditions and diverse biological pathways, influencing cellular proliferation, programmed cell death, immune response, and metabolic regulation [17,18]. In women, the primary circulating estrogens include estradiol, estrone, and estrone sulfate, shifting the balance towards estrone during menopause. Both 17β-estradiol (E2) and estrone are associated with menopausal symptoms, and research has shown that decreased E2 production in ovariectomized experimental animals accelerates the progression of renal disease [19]. Furthermore, E2 supplementation has been shown to protect against kidney damage by influencing pathways such as inflammation and collagen synthesis [20]. Despite its known protective roles, the detailed mechanisms by which E2 influences kidney injury remain unclear, providing a basis for further investigation into its therapeutic potential in CKD.

Based on previous reports, we aimed to understand the sex differences in an adenine diet (AD)-induced kidney fibrosis model. AD-induced kidney fibrosis is accompanied by severe inflammatory responses in tubular epithelial cells, leading to fibroblast activation and fibrosis. We evaluated the extent of kidney function, structural kidney damage, and fibrosis by histological and biochemical analyses. We further examined the pro-inflammatory responses in the same mouse model. Based on these observations, we further investigated the potential anti-inflammatory and anti-fibrotic effects of E2 both in vivo and in vitro.

## 2. Results

### 2.1. Male Mice Exhibit Heightened Vulnerability to AD-Induced Kidney Injury

To evaluate the impact of AD-induced kidney damage and its relationship to sex differences, male and female mice were fed 0.25% AD for two weeks, while control groups received a standard chow diet (Figure 1A). Both male and female mice exhibited significant weight loss following AD feeding, but kidney weights remained stable (Figure 1B,C). The kidney weight-to-body weight ratio, however, increased significantly in AD-fed mice (Figure 1D). Next, kidney function was evaluated, revealing that basal levels of BUN and creatinine were significantly elevated in AD-fed mice (Figure 1E,F). AD-fed female mouse kidneys had lower BUN levels than AD-fed male mice kidneys, however, no difference in serum creatinine levels was observed (Figure 1E,F). Structural changes were measured using H&E staining. In male mice, AD led to severe tubular dilation, cell loss, and cast formation, whereas female mice showed only mild injuries (Figure 2A,B). Immunostaining for kidney injury molecule-1 (KIM-1), a biomarker of renal proximal tubule damage, revealed higher expression in AD-fed male kidneys compared to females (Figure 2D). Similarly, AD-fed female mice exhibited lower mRNA expression levels of kidney injury markers, including *Havcr1* and *Ssr1*, relative to male kidneys (Figure 2C). These results indicated that female mice were less vulnerable to AD-induced kidney damage.

### 2.2. Male Mice Exhibited Severe Renal Fibrosis Development in the AD Model

Fibrosis progression was compared using the same experimental model. To compare fibrosis development between sexes, we analyzed ECM-related gene and protein expression. *Col1a2*, *Vim*, and *Tgfb* gene expression increased in both male and female kidneys after AD feeding but was significantly higher in males (Figure 3A). Protein expression of α-SMA and COL1 were also elevated in AD-fed male kidneys, consistent with the gene expression data (Figure 3B). Sirius Red staining and immunohistochemistry for α-SMA were confirmed to evaluate fibrosis histologically, revealing that AD-fed male mice displayed a larger Sirius Red-positive area compared to females (Figure 3C). However, only mild changes were observed in the female mice (Figure 3C,D). Similar results were observed when α-SMA was detected by immunohistochemistry (Figure 3E). These data suggested that the kidneys of male mice developed severe fibrosis after AD administration.

### 2.3. Male Mice Are Susceptive to Inflammatory Responses in the Kidney Under AD Conditions

Given the link between inflammation and fibrosis [21], we compared the inflammatory response in AD-fed male and female mouse models. The kidney expression levels of pro-inflammatory (*Tnfa*, *Il6*, and *Ccl2*) and macrophage marker genes (*Emr1* and *Cd68*) were upregulated in both sexes, with higher expression in males (Figure 4A,B). We next investigated the expression and phosphorylation of NF-κB. Both male and female kidneys exhibited increased expression and phosphorylation levels of p65 in response to AD administration, though males showed a more substantial response (Figure 4C). Expression of *Ccl2* and *Emr1* was histologically confirmed using double ISH analysis. In male mice, AD kidneys showed a higher expression of *Ccl2* in the damaged tubules and interstitial regions after AD feeding than in female mice (Figure 4D). Additionally, *Emr1*-expressing cells were present near *Ccl2*-expressing cells and were significantly increased in AD-fed male mice (Figure 4D). These data indicated that pro-inflammatory responses and macrophage infiltration were highly elevated in AD-induced male mice kidneys, accompanied by severe fibrosis.

### 2.4. E2 Shows Protective Effects on LPS-Induced Inflammatory Responses in Renal Tubular Epithelial Cells

In previous studies, we and others reported that renal epithelial cells influence macrophage infiltration by modulating chemokine expression during fibrosis development [12,22,23]. To explore whether E2 regulates inflammatory responses in kidney epithelial cells, we treated NRK52E cells with LPS (Figure 5A). LPS significantly increased chemokine gene expression (*Ccl2*, *Ccl3*, and *Cxcl1*), whereas the E2 pre-treatment markedly showed a significant decrease in their expression (Figure 5B). We further tested whether E2 inhibits LPS-induced NF-κB transcriptional activity. Luciferase assays showed that E2 reduced NF-κB transcriptional activity in LPS-treated cells (Figure 5C). Furthermore, E2 treatment effectively suppressed nuclear translocation of NF-κB p65 in LPS-treated cells (Figure 5D). Consistently, Western blot analysis showed reduced p65 nuclear translocation and phosphorylation in E2-treated cells (Figure 5E). These findings suggest that E2 reduced LPS-induced inflammation by inhibiting NF-κB activation in NRK52E cells.

### 2.5. E2 Decreases TGFβ-Induced Fibroblast Activation in NRK49F Cells

To demonstrate the direct anti-fibrotic effects of E2, we utilized kidney fibroblasts treated with TGFβ (Figure 6A). In NRK49F cells, TGFβ1 markedly increased *Col1a2*, *Acta2*, *Vim*, and *Col3a1* transcript levels, whereas E2 pre-treatment significantly reduced the expression of those genes (Figure 6B). We further evaluated the protein expression of VIM, α-SMA, p-SMAD2, and p-SMAD3, crucial transcription factors in the transcriptional regulation of ECM-related genes. E2 pre-treatment effectively decreased the expression of these proteins in TGFβ-treated fibroblasts (Figure 6C). Similar results were observed when VIM and α-SMA were detected using immunofluorescence staining (Figure 6D,E). These results suggest that E2 directly decreases TGFβ1-induced fibroblast activation by inhibiting SMAD2/3 activity.

### 2.6. E2 Treatment Ameliorates AD-Induced Kidney Inflammation and Fibrosis

The protective effects of E2 were further validated in vivo using AD-fed male mice. E2 was orally administered at a 0.1 mg/kg/day dose throughout the experiment (Figure 7A). We confirmed that AD consumption reduced the body and kidney weight of the mice (Figure 7B). However, E2 had no effect on these parameters (Figure 7B). Serum BUN levels were significantly elevated in AD-fed mice, however, the E2 group showed reduced BUN levels (Figure 7C). Histological analysis using H&E and Sirius Red staining showed that E2 mitigated kidney damage, as detected by tubular dilation, cell loss, immune cell infiltration, and fibrosis in AD-fed mice (Figure 7D–F). Western blot analysis demonstrated that E2 reduced the expression of fibrosis-associated proteins and reduced NF-κB activation (Figure 7G,H). Immunohistochemical staining also showed increased positive staining for p-p65 in the tubular cells of kidneys from AD-fed mice, but there was a decrease in the positive area of p-p65 staining after E2 treatment (Figure 7I). These observations indicated that E2 significantly alleviated inflammation and fibrosis in AD-induced kidney damage.

## 3. Discussion

In this study, we demonstrated the sex-specific differences in AD-induced kidney toxicity and the protective effects of E2 on renal inflammation and fibrosis. Our results indicated the pronounced susceptibility of male mice to AD-induced kidney damage, characterized by increased BUN levels, inflammatory responses, and fibrosis, compared to female mice. This heightened susceptibility in male mice is likely due to lower endogenous estrogen levels, which confer protection against renal injury. Through both in vitro and in vivo experiments, we elucidated the anti-inflammatory and anti-fibrotic effects of E2. In NRK52E cells, E2 pre-treatment significantly reduced LPS-induced inflammatory responses by inhibiting NF-κB activation and decreasing chemokine expression. Similarly, in NRK49F renal fibroblasts, E2 attenuated TGFβ-induced fibrotic responses, as evidenced by reduced expression of ECM-related genes and proteins. In vivo studies further supported these findings, showing that E2 treatment notably reduced fibrosis and inflammation in AD-fed male mice. These results suggest that estrogen confers renal protection and highlights the dual anti-inflammatory and anti-fibrotic roles of E2, proposing its potential therapeutic application in addressing sex differences in CKD progression.

Sex disparities significantly influence CKD progression, as evidenced by both clinical and experimental studies [24,25]. Epidemiological evidence across multiple populations indicates that men are disproportionately affected by ESRD, with a lifetime risk estimated to be nearly 50% higher than that observed in women. Furthermore, recent longitudinal studies suggest that women typically experience slower CKD progression and reduced mortality rates compared to their male counterparts [6,26,27]. Experimental models have consistently demonstrated that male animals experience an accelerated progression of kidney diseases [28,29]. Our research supports these findings by revealing structural changes and declines in kidney function in male rodents subjected to AD-induced kidney injury [12]. However, the protective effects observed in females appear to diminish after menopause. The mechanisms responsible for this protective effect in younger women are not yet fully understood but are thought to involve ovarian steroids, which play key roles in modulating inflammation, oxidative stress, blood pressure homeostasis, and processes like cellular apoptosis and regeneration [30,31]. Moreover, the experimental manipulation of hormones in CKD models has replicated these sex-specific effects, suggesting that female sex hormones, particularly estrogens, could potentially slow the progression of CKD, whereas male hormones may aggravate it. Therefore, interventions aimed at restoring ovarian steroids or their mimics in postmenopausal women and those with low estrogen levels could potentially mitigate renal damage pathways and delay or prevent the progression to ESRD.

Estrogen, primarily synthesized in the ovaries, is essential for female reproductive health, while in males, testosterone is converted into E2 by aromatase [32]. Among the various forms of estrogens, E2 is the most biologically active and has been shown to confer significant renoprotection. However, the effects of E2 on kidney injury remain unclear. Several studies have demonstrated the beneficial effects of E2. One study found that a decrease in E2 synthesis in experimental animals following ovariectomy (OVX) accelerates renal disease progression by affecting the renin-angiotensin system’s activity [33]. Another study observed that E2 ameliorated renal injury by decreasing tubular cell apoptosis in models of acute aristolochic acid nephropathy, possibly by inhibiting p53 signaling [34]. Furthermore, in an aging rat model, E2 attenuated OVX-induced glomerulosclerosis and tubulointerstitial fibrosis [35]. However, some studies have reported conflicting results. Pezashki et al. found that estrogen does not protect nephrons against cisplatin-induced oxidative stress or nephrotoxicity [36]. Our study found that E2 administration in AD-fed male mice significantly alleviated renal fibrosis and inflammation, primarily the inhibition of NF-κB and SMAD2/3 pathway activity. These results provide strong evidence supporting the hypothesis that estrogen confers renal protection.

Our results revealed significant inflammation in the AD-induced kidney fibrosis models. Adenine is known to cause direct damage to tubular epithelial cells (TECs), elevates inflammatory responses, and promotes fibrosis. In kidney damage, TECs actively contribute to inflammation via chemokine production. Multiple studies have indicated that these chemokines are crucial for recruiting monocytes and macrophages, which play pivotal roles in pro-inflammatory kidney diseases characterized by their accumulation [12,37]. A key player in this process is NF-κB, a major regulator of inflammation and is activated in kidney injury, leading to increased chemokine and cytokine production. Enhanced activation of NF-κB has been identified in TECs under conditions of kidney injury and exposure to inflammatory signals [38]. Notably, studies have demonstrated that selective inhibition of NF-κB in TECs mitigates renal damage following ischemic injury by decreasing chemokine production and limiting inflammatory cell infiltration [38]. Similarly, suppressed NF-κB expression in kidney tissue has been found to alleviate renal fibrosis induced by folic acid in mouse models [39]. Given these observations, modulation of epithelial inflammation has emerged as a promising target for controlling kidney inflammation and fibrosis. In our study, E2 treatment led to the suppression of NF-κB nuclear translocation and transcriptional activity, reducing chemokine production in epithelial cells. These mechanisms underlie the protective role of E2 against renal inflammation in female mice and E2-treated male mice.

Kidney fibrosis, a key feature of CKD progression, involves the loss of functional renal cells and their subsequent replacement with ECM components. In the fibrotic process, both ECM synthesis and degradation increase as part of tissue remodeling, however, the rate of ECM production exceeds its breakdown, leading to excessive accumulation [40]. As fibrosis progresses, fibroblasts transform into myofibroblasts, leading to substantial ECM protein production and significantly contributing to the kidney’s fibrotic phenotype [41]. TGF-β is a key regulator of renal fibrosis, primarily by driving the activation of kidney fibroblasts. SMAD3, a critical downstream mediator of TGFβ1 signaling, promotes ECM production and fibrosis progression by directly interacting with SMAD-binding elements (SBE) located in the promoter regions of target genes via its MH1 domain [42,43]. Essential SMAD3 target genes implicated in TGFβ1-induced ECM deposition include *Ctgf*, *Col1a2*, *Col2a1*, *Col3a1*, *Acta2*, and *Vimentin* [44]. In this study, we observed that E2 treatment mitigated TGF-β-induced fibrotic responses by inhibiting SMAD2/3 activity and enhancing ECM protein production in fibroblasts.

Recent studies have shown that tamoxifen, a selective estrogen receptor modulator (SERM), effectively reduced renal tubular injury and fibrosis induced by unilateral ureteral obstruction (UUO) [45,46]. This effect is attributed to Tamoxifen and has been shown to inhibit TGFβ1-induced fibroblast proliferation and migration through the modulation of the estrogen receptor (ER) α-dependent TGF-β1/SMAD signaling pathway in vitro [46]. Additionally, research by Cao et al. suggests that increased TGFβ/SMAD3 signaling may lead to renal ERβ depletion and development of renal fibrosis in CKD human and mouse models [47]. The mechanism by which estrogen potentially inhibits these transcription factors likely involves the molecular actions of ER. As a nuclear receptor, ERβ directly interacts with the SMAD3, preventing its association with SBE sequences in the promoters of downstream genes [47]. This interaction suppresses the transcription of profibrotic genes. Despite ERβ being recognized as the primary estrogen receptor subtype mediating estrogen’s effects in the kidney, its expression in fibroblasts remains unclear. Future investigations should explore whether E2 inhibits TGFβ/SMAD3 signaling through ERβ and elucidate ERβ’s role in fibroblasts. These findings underscore the potential therapeutic implications of SERMs like Tamoxifen and highlight the importance of understanding ERβ’s function in renal fibrosis for developing targeted treatments for chronic kidney diseases.

While our study offers valuable insights into sex differences in AD-induced kidney injury, it also raises several questions and acknowledges its limitations. First, a single dose of E2 may not fully encompass the spectrum of physiological responses possible with varying doses. Moreover, this study did not evaluate the long-term effects of the E2 treatment. Future research employing diverse doses and extended treatment durations is essential to comprehensively understand the chronic effects of E2 on kidney health and its potential recovery mechanisms. Although our mouse models provide valuable insights, they may not entirely replicate the human CKD pathology. Therefore, translational studies involving human subjects are crucial for validating these findings in a clinical context. Despite these limitations, our study highlights the significant sex disparities in AD-induced kidney toxicity, demonstrating that male mice are more susceptible to kidney damage and fibrosis. The protective effects of E2 highlight its potential as a therapeutic option, particularly in males with a higher risk of CKD progression. These findings emphasize the importance of considering sex differences in the development of CKD management strategies and pave the way for future investigations into sex-specific treatments for kidney diseases.

## 4. Materials and Methods

### 4.1. Animal Studies

All animal experiments complied with the Institutional Animal Care Committee guidelines at Pusan National University (No. PNU-2022-0238, 1 November 2022, PNU-IACUC). We acquired 12-week-old C57BL/6J mice (Hyochang Science, Daegu, Republic of Korea) and fed both male and female groups a diet containing 0.25% adenine (TCI; A0149) over 14 days to induce kidney injury (*n* = 5–6 per group). Additionally, to investigate the effect of E2 on kidney injury, a separate group of male mice received daily oral administration of E2 (Sigma-Aldrich, St. Louis, MO, USA; 3301) at a dose of 0.1 mg/kg/day in corn oil for 14 days. All animals were kept at a temperature of 23 ± 2 °C with 60 ± 5% relative humidity on 12-h light/dark cycle. Serum and kidney tissue samples were subsequently collected for biochemical analyses.

### 4.2. Cell Culture and Treatments

Rat renal tubular epithelial cells (NRK52E) and rat renal fibroblasts (NRK49F) were obtained from the ATCC (Manassas, VA, USA). Cells were cultured in DMEM containing 5% or 10% fetal bovine serum. In NRK52E cells, E2 (100 nM) was treated 0.5 h prior to lipopolysaccharide (LPS, 10 μg/mL) treatment for 1 h to examine NF-κB activation. In NRK49F cells, E2 pre-treatment (100 nM) occurred 0.5 h before treating TGFβ (10 ng/mL) for 24 h to evaluate ECM production.

### 4.3. Biochemical Assays for BUN and Creatinine

Blood urea nitrogen (BUN) and creatinine levels were measured in the mouse serum using a kit from Shinyang Diagnostics (Seoul, Republic of Korea) following the manufacturer’s protobio-rad

### 4.4. Protein Extraction and Western Blot Analysis

Kidney and cellular proteins were lysed as previously described with slight modifications [48]. Primary antibodies were as follows; α-SMA (sc32251, Santa Cruz, Dallas, TX, USA), COL1 (sc80760, Santa Cruz), α-tubulin (sc8035, Santa Cruz), p-NF-κB (sc101749l, Santa Cruz), NF-κB (sc372, Santa Cruz), LaminA/C (sc376248, Santa Cruz), β-actin (sc69879, Santa Cruz), p-SMAD3 (#9520, Cell Signaling Technology, Danvers, MA, USA), Vimentin (#5741S, Cell Signaling Technology), and p-SMAD2 (ab184557, Abcam, Cambridge, MA, USA).

### 4.5. RNA Extraction and Quantitative Polymerase Chain Reaction (qPCR)

Total RNA was isolated from kidney tissues or cells using TRIzol reagent (Invitrogen, Carlsbad, CA, USA). Total RNA was reverse transcribed using a SuPrimeScript cDNA synthesis kit (GENETBIO, Deajeon, Republic of Korea). qPCR was conducted with the SYBR green master mix (GENETBIO) on a CFX96 Connect System (Bio-Rad, Hercules, CA, USA). Primers were designed with the Primer3Plus program [49] and are listed in Appendix A.

### 4.6. Histological Analysis

Briefly, kidneys were fixed in 10% neutral formalin, embedded in paraffin, and sectioned for Hematoxylin and Eosin (H&E) staining. For assessing renal fibrosis, Sirius Red (SR) staining was performed with a Vitrovivo kit (VB-3017, Rockville, MD, USA). The percentage of positive staining area was analyzed using ImageJ software v1.8.0. For immunohistochemistry, paraffin sections (5 μm) were incubated with a primary antibody specific for α-SMA (sc32251, Santa Cruz) and visualized with a diaminobenzidine substrate, followed by counterstaining with hematoxylin. For fluorescence staining, the paraffin-embedded kidney sections were incubated with primary antibodies KIM1 (ab47635, Abcam), NFκB (sc372, Santa Cruz), Vimentin (#5741S, Cell signaling, Danvers, MA, USA), and α-SMA (sc32251, Santa Cruz). Secondary antibody incubation was carried out with VectaFluor Duet Reagent, consisting of DyLight 488 Anti-Rabbit IgG and DyLight 594 Anti-Mouse IgG cocktails in the kit (8818, Vector Laboratories, Burlingame, CA, USA). Nuclear staining was performed with DAPI to visualize cell nuclei. Images were captured by LS30 microscope (Leam solution, Seoul, Republic of Korea).

### 4.7. Quantification of Dilated Tubules

The tubular injury was scored using H&E staining and was overlaid on a grid measuring 13.625 mm [50]. Next, we assessed the dot count within the lumen to evaluate the extent of tubule dilation based on the following established criteria: (1) normal tubules were represented by one dot, (2) dilated tubules by two dots, (3) microcysts by three to ten dots, and (4) cysts by more than ten dots.

### 4.8. In Situ Hybridization (ISH) Staining

ISH was conducted on paraffin-embedded kidney tissue sections using RNAscope 2.5 HD Duplex Detection Kit (322436, Biotechne, Minneapolis, MN, USA). Probes targeting specific mRNAs, such as Mm-Emr1 (#317969-C2) and Mm-Ccl2 (#311791), were used for capturing detailed images of tissue structure using an LS30 microscope.

### 4.9. Quantification and Statistical Analysis

Differences between the two groups were assessed using Student’s t-test, while intergroup variations were analyzed through analysis of variance (ANOVA). A p-value of less than 0.05 was considered statistically significant. Statistical calculations were conducted using GraphPad Prism version 5 (GraphPad Software Inc., San Diego, CA, USA). Image analyses were quantified using ImageJ software v1.8.0.

## Figures and Tables

**Figure 1 ijms-26-01358-f001:**
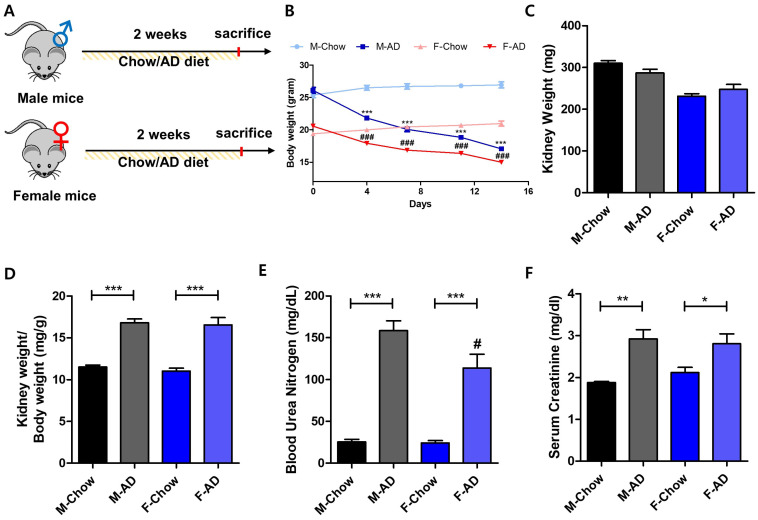
Comparison of gender differences in kidney toxicity induced by adenine diet. (**A**) Experimental workflow illustrating the adenine diet (AD)-induced kidney injury model. (**B**) Changes in body weight of male and female mice during the experimental period. *** *p* < 0.001 vs. Chow diet-fed male mouse kidney groups. ### *p* < 0.001 vs Chow diet-fed female mouse kidney groups. (**C**) Kidney weights in AD-treated groups. (**D**) The ratio of kidney weight/body weight. *** *p* < 0.001 vs. two groups indicated. (**E**) Blood urea nitrogen levels in AD-induced kidney injury model. Statistical comparisons: *** *p* < 0.001. # *p* < 0.05 vs. AD-fed male mouse kidney groups. (**F**) Serum creatinine concentration in AD-fed mouse kidneys, with ** *p* < 0.01 and * *p* < 0.05 marking significance.

**Figure 2 ijms-26-01358-f002:**
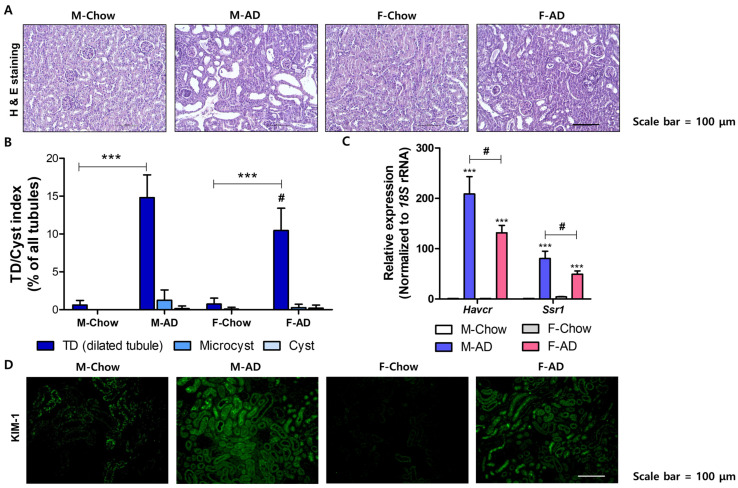
Male mice were more susceptive to AD-induced kidney damages. (**A**) Representative H&E-stained sections showing kidney morphology in AD-fed and control groups. Scale bars: 100 μm. (**B**) Quantitative analysis of renal injury scores in each group. *** *p* < 0.001 vs. two groups indicated. # *p* < 0.05 vs. AD-fed male mouse kidney groups. (**C**) Relative mRNA expression levels for *Havcr1* and *Ssr1* from each kidney. *** *p* < 0.001, vs. control groups. # *p* < 0.05 between the two groups indicated. (**D**) Immunofluorescence image of kidney injury molecule 1 (KIM-1) (green) in kidney sections, with scale bars = 100 μm.

**Figure 3 ijms-26-01358-f003:**
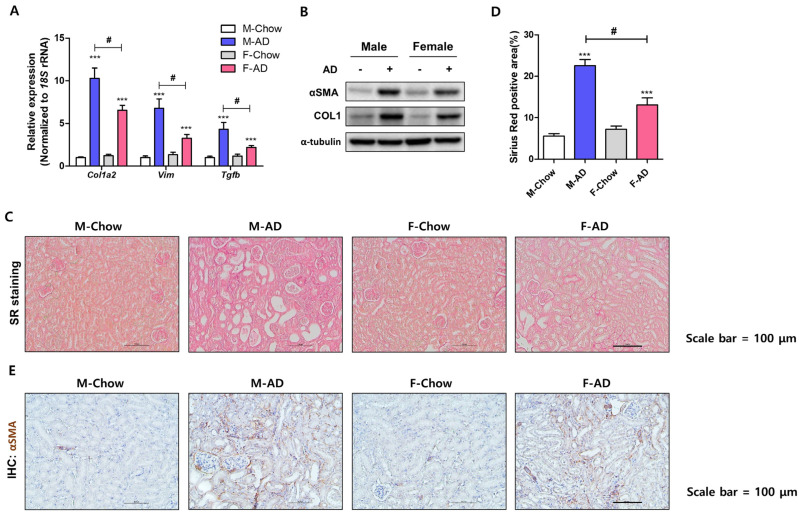
AD feeding induced more severe fibrosis in male mouse kidneys. (**A**) qRT-PCR of mRNA for *Col1a2*, *Vim*, and *Tgfb* in male and female kidneys after AD administration. *** *p* < 0.001 vs. control groups. # *p* < 0.05 between the two groups indicated. (**B**) Western blot results showing α-SMA and COL1 protein levels, with α-tubulin as the loading control. (**C**) Sirius Red staining of kidney sections from control and AD-fed mice, with scale bars = 100 μm. (**D**) Quantification of the percentage area fibrosis with Sirius Red staining. *** *p* < 0.001, vs. control groups. # *p* < 0.05 between the two groups indicated. (**E**) Immunohistochemical staining of α-SMA to assess fibrosis distribution. Scale bars = 100 μm.

**Figure 4 ijms-26-01358-f004:**
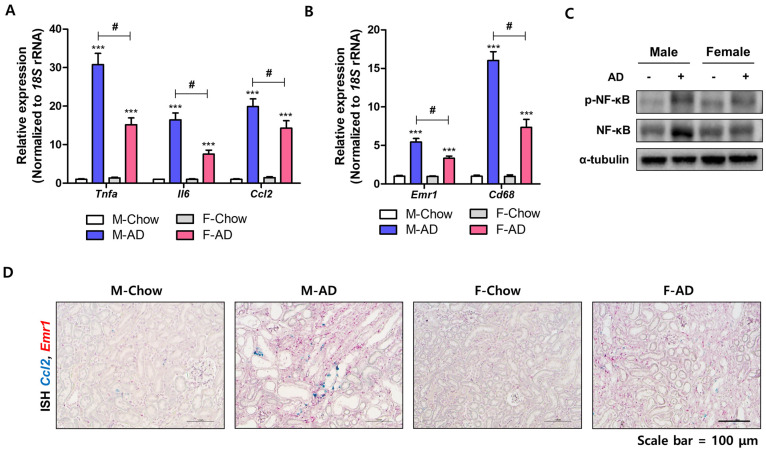
The AD-induced inflammatory response was more pronounced in male kidneys than in female kidneys. (**A**) qRT-PCR was performed for *Tnfa*, *Il6*, and *Ccl2* mRNA expressions in male and female AD-fed kidneys. *** *p* < 0.001 vs. control groups. # *p* < 0.05 between the two groups indicated. (**B**) Expression of macrophage *Cd68* and *Emr1* genes in AD-fed kidneys. *** *p* < 0.001 vs. control groups. # *p* < 0.05 between the two groups indicated. (**C**) Western blot results showing NF-κB and *p*-NF-κB levels, normalized to α-tubulin. (**D**) In situ hybridization images displaying *Ccl2* (green) and *Emr1* (red) mRNA localization in kidney sections. Scale bars = 100 μm.

**Figure 5 ijms-26-01358-f005:**
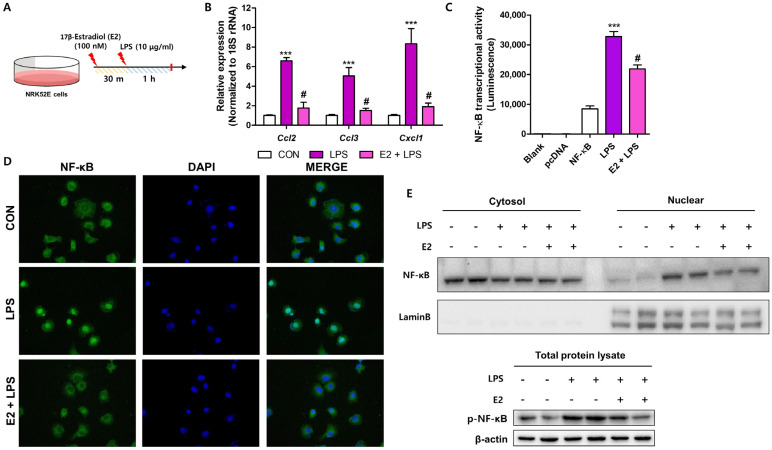
E2 exhibited protective effects against LPS-induced inflammatory responses in NRK52E cells. (**A**) Overview of LPS and E2 treatment protocol for NRK52E cells. (**B**) Changes in pro-inflammatory gene expression (*Ccl2*, *Ccl3*, and *Cxcl1*) following treatments. *** *p* < 0.001 vs. control group. # *p* < 0.05 vs. LPS group. (**C**) Luciferase assay showing NF-κB activity under different conditions. *** *p* < 0.001 vs. NF-κB luciferase vector treated group. # *p* < 0.05 vs. LPS treated group. (**D**) Immunofluorescence images of NF-κB (green) in the NRK52E cells, with scale bars = 100 μm. (**E**) Western blot analysis showing cytosolic and nuclear fractions of NF-κB, with p65 phosphorylation and appropriate loading controls (LaminB for nuclear fraction, β-actin for cytosolic fraction).

**Figure 6 ijms-26-01358-f006:**
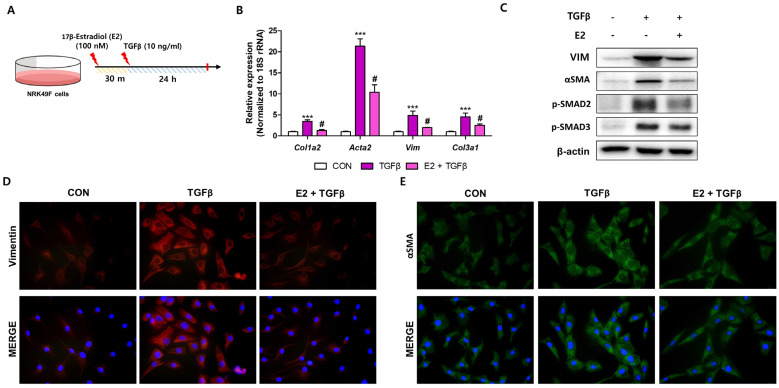
E2 treatment showed anti-fibrotic effects in NRK49F fibroblasts. (**A**) Experimental scheme of TGFβ treatment in NRK49F cells. (**B**) Relative expression levels of ECM-related genes (*Col1a2*, *Acta2*, *Vim*, and *Col3a1*) after treatments. *** *p* < 0.001 vs. control group. # *p* < 0.05 vs. TGFβ treated group. (**C**) Protein levels of VIM, αSMA, p-SMAD2, and p-SMAD3 in NRK49F cells, with β-actin as the loading control. (**D**) Immunofluorescent staining of the VIM antibody (red) in NRK49F cells, with scale bars = 100 μm. (**E**) Immunofluorescence images with the αSMA antibody (green) in NRK49F cells, with scale bars = 100 μm.

**Figure 7 ijms-26-01358-f007:**
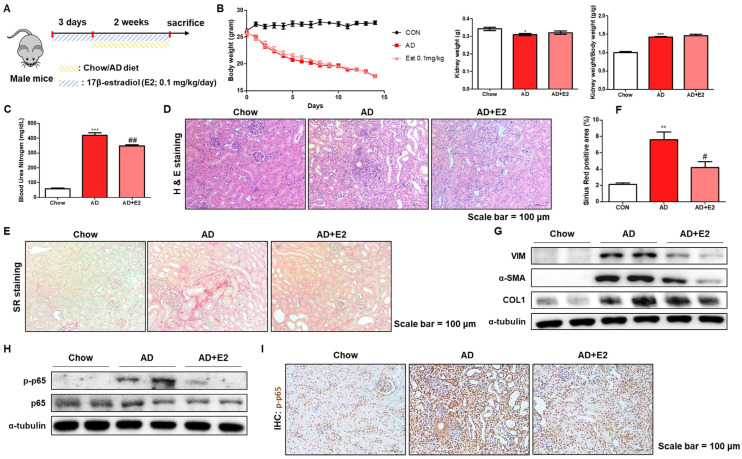
E2 treatment resulted in down-regulated renal inflammation and fibrosis induced by AD administration. (**A**) Experimental design of E2 treatment in AD-fed mouse models. (**B**) Body, kidney, and kidney/body weight ratios in each group. * *p* < 0.05 and *** *p* < 0.001 vs. Chow diet-fed male mice groups. (**C**) Changes in serum BUN levels in study groups. *** *p* < 0.001 vs. Chow diet-fed male mice groups. ## *p*< 0.01 vs. AD-fed mice groups. (**D**) Results of H&E staining for kidney sections. (**E**) Representative images of Sirius Red staining in kidney sections. Scale bars = 100 μm. (**F**) The positive area of SR staining was quantified for the study groups. Scale bars = 100 μm. ** *p* < 0.01 vs. Chow diet-fed male mice groups. # *p* < 0.05 vs. AD-fed mice groups. (**G**) Western blot analysis of fibrosis markers (VIM, α-SMA, and COL1) and (**H**) NF-κB activity (p-p65, p65). (**I**) Immunohistochemistry showing p-p65 expression patterns in the experimental groups. Scale bars = 100 μm.

## Data Availability

Data are contained within the article.

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
