# Peer review of "Gender Differences in Adenine Diet-Induced Kidney Toxicity: The Impact of 17β-Estradiol on Renal Inflammation and Fibrosis"

_ijms, 2025, doi:10.3390/ijms26031358_

Round 1
Reviewer 1 Report
Comments and Suggestions for Authors
The authors present an interesting paper evaluating the effect of estradiol on chronic kidney damage in an animal model.
Both in vivo and in vitro evidence of the effect of estradiol on the development of kidney damage is presented.
The results adequately demonstrate estradiol's protective effect on kidney damage.
However, for this article to be accepted for publication, there are minor issues that must be clarified
1) The authors analyzed specific molecules such as Havcr, Ssr1, Kim-1 Vim1, and the results showed decreased expression with estradiol treatment. However, in the introduction, nothing is mentioned about the signaling pathways they participate in and why these analyses are essential for CKD . In the discussion section, again, the authors don't discuss how the treatment with estradiol modifies their expression molecules.
2) The "animal studies" section in Material and Methods is confusing since it mentions that males and females were used and administered Adenine to develop kidney damage. All were treated daily with estradiol (male and female). However, it only shows the results of males treated with estradiol. It is important to clarify this point.
3) Figure 1B shows the weight of the animals before and after two weeks of kidney damage. I recommend showing the variation in weight over time (1 week, 2 weeks) to determine whether the effect of estradiol is early or late.
Author Response
Reviewer 1.
Comments 1: The authors analyzed specific molecules such as Havcr, Ssr1, Kim-1 Vim1, and the results showed decreased expression with estradiol treatment. However, in the introduction, nothing is mentioned about the signaling pathways they participate in and why these analyses are essential for CKD. In the discussion section, again, the authors don't discuss how the treatment with estradiol modifies their expression molecules.
Response 1: Thank you for your comment. While we analyzed specific molecules such as Havcr, Ssr1, Kim-1, and Vim1, we did not specifically mention Kim-1 (=Havcr1) and Vimentin (=Vim) in the Introduction because Kim-1 is a well-established marker routinely used to assess kidney damage, and Vimentin, like collagens and α-SMA, is a representative marker of fibrosis-associated ECM accumulation. Furthermore, these molecules were briefly explained in the Results section (page.2, line 48-50 in Introduction/page.3, line 101-103 in Result section).
Comments 2: The "animal studies" section in Material and Methods is confusing since it mentions that males and females were used and administered Adenine to develop kidney damage. All were treated daily with estradiol (male and female). However, it only shows the results of males treated with estradiol. It is important to clarify this point.
Response 2: Thank you for pointing this out. We agree with this comment. Therefore, we have revised the manuscript to clarify this point accordingly (page.11, The “animal studies” section in Materials and Methods).
Comments 3: Figure 1B shows the weight of the animals before and after two weeks of kidney damage. I recommend showing the variation in weight over time (1 week, 2 weeks) to determine whether the effect of estradiol is early or late.
Response 3: Thank you for your valuable suggestion. We have revised Figure 1B and Figure 7B to include weight variations over time to better determine whether the effect of estradiol is early or late (Figure 1 and 7).

Reviewer 2 Report
Comments and Suggestions for Authors
The article is well-written and the discussion supports the experimental findings. The limitations of this study are clearly acknowledged. I have 3 additional comments to the investigators:
1. Hemodynamic parameters may differ between males and females and these differences may promote the fibrotic and inflammatory processes. Please add data for hemodynamic parameters in your analysis.
2. Statistical analyses included several comparisons, raising the possibility for bias due to multiple hypothesis testing. Please provide some statistical adjustments to certify that significant between-group differences are not a matter of chance.
3. Please provide some numerical data in the abstract.
Author Response
Reviewer 2.
Comments 1: Hemodynamic parameters may differ between males and females and these differences may promote the fibrotic and inflammatory processes. Please add data for hemodynamic parameters in your analysis.
Response 1: Thank you for your insightful comment. Hemodynamic parameters such as blood flow and pressure require additional experimental measurements, which would necessitate a new set of animal experiments. Unfortunately, it is not feasible to conduct these additional experiments at this stage. However, we recognize the importance of this aspect and will consider incorporating hemodynamic analyses in our future studies.
Comments 2: Statistical analyses included several comparisons, raising the possibility for bias due to multiple hypothesis testing. Please provide some statistical adjustments to certify that significant between-group differences are not a matter of chance.
Response 2: Thank you for pointing this out. We agree with this comment. To address this, we reanalyzed the data using ANOVA, and the results remained consistent with our previous findings. We have attached the updated statistical analysis as a PDF file for your review.
Comments 3: Please provide some numerical data in the abstract.
Response 3: Thank you for your suggestion. We have revised the abstract to include numerical data as requested (page.1, The Abstract section).
